# Establishing a Prediction Model for the Efficacy of Platinum—Based Chemotherapy in NSCLC Based on a Two Cohorts GWAS Study

**DOI:** 10.3390/jcm12041318

**Published:** 2023-02-07

**Authors:** Qi Xiao, Chenxue Mao, Ying Gao, Hanxue Huang, Bing Yu, Lulu Yu, Xi Li, Xiaoyuan Mao, Wei Zhang, Jiye Yin, Zhaoqian Liu

**Affiliations:** 1Department of Clinical Pharmacology, Xiangya Hospital, Central South University, 87 Xiangya Road, Changsha 410008, China; 2Institute of Clinical Pharmacology, Central South University, Hunan Key Laboratory of Pharmacogenetics, 110 Xiangya Road, Changsha 410078, China; 3National Clinical Research Center for Geriatric Disorders, 87 Xiangya Road, Changsha 410008, China; 4Institute of Clinical Pharmacology, Engineering Research Center for Applied Technology of Pharmacogenomics of Ministry of Education, Central South University, Changsha 410078, China; 5Department of Geriatric Respiratory and Critical Care Medicine, Xiangya Hospital, Central South University, Changsha 410008, China

**Keywords:** platinum, NSCLC, efficacy, model, SNP

## Abstract

Platinum drugs combined with other agents have been the first-line treatment for non-small cell lung cancer (NSCLC) in the past decades. To better evaluate the efficacy of platinum–based chemotherapy in NSCLC, we establish a platinum chemotherapy response prediction model. Here, a total of 217 samples from Xiangya Hospital of Central South University were selected as the discovery cohort for a genome-wide association analysis (GWAS) to select SNPs. Another 216 samples were genotyped as a validation cohort. In the discovery cohort, using linkage disequilibrium (LD) pruning, we extract a subset that does not contain correlated SNPs. The SNPs with *p* < 10^−3^ and *p* < 10^−4^ are selected for modeling. Subsequently, we validate our model in the validation cohort. Finally, clinical factors are incorporated into the model. The final model includes four SNPs (rs7463048, rs17176196, rs527646, and rs11134542) as well as two clinical factors that contributed to the efficacy of platinum chemotherapy in NSCLC, with an area under the receiver operating characteristic (ROC) curve (AUC) of 0.726.

## 1. Introduction

Platinum is one of the most promising and widely used drugs for the treatment of various cancers. Cisplatin and other platinum–derivative drugs exert anticancer activity mainly by binding to DNA and forming adducts that affect DNA transcription and replication [1]. However, in many cases, tumor cells exposed to platinum can activate a multi-mechanism adaptive response that leads to drug resistance, which limits the clinical use of platinum drugs. Resistance to cisplatin depends on multiple factors such as reduced drug accumulation, inactivation of the drug by binding to non-target proteins, increasing DNA repair, and altering of signals to apoptosis [2,3,4,5].

Non-small cell lung cancer (NSCLC) accounts for approximately 85% of all lung cancers and is the major subtype of lung cancer [6]. Platinum drugs are used as first-line agents in the clinical practice guidelines for NSCLC. Unfortunately, patients differ greatly in their responses to platinum drugs [7]. Therefore, it is helpful for clinical treatment to study the efficacy of platinum in NSCLC. We hope to find pharmacogenomic markers that influence the efficacy of platinum chemotherapy, and to further develop models to predict the efficacy of platinum chemotherapy.

In one of our previous studies, a predictive model was established for platinum–based chemotherapy response, with a sensitivity of 0.90, specificity of 0.47, and area under the receiver operating characteristic (ROC) curve (AU-ROC) of 0.80 [8]. By applying this model to clinical practice, we found that its accuracy needs to be further improved, and the number of detected SNPs needs to be further optimized and reduced.

This study intends to improve on previous studies by adopting GWAS to find more genetic factors affecting the efficacy of NSCLC platinum chemotherapy. BayesNet, NaiveBayes, Random Forest (RF), Logistic, support vector machine (SVM), Bagging, Decision Tree (DT), K-Nearest-Neighbors (KNN), Random Tree (RT), AdaBoost, and Polygenic risk score (PRS), will be used to establish the new model. After that, the previous model will be validated in another cohort, and finally, clinical factors will be incorporated to optimize the model.

## 2. Materials and Methods

### 2.1. Study Population and Clinical Data Collection

The patients in this study were recruited from Xiangya Hospital of Central South University and Hunan Cancer Hospital (Changsha, China), from 2012 to 2019, and were diagnosed with NSCLC by histopathological examination. The patients received platinum–based chemotherapy regimens for at least 2 cycles: one course of treatment lasts for three weeks. The platinum–based drugs used were cisplatin (75 mg/m^2^) or carboplatin (AUC 5) on day 1, in combination with pemetrexed (500 mg/m^2^) on day 1, gemcitabine (1250 mg/m^2^) on days 1 and 8, paclitaxel (175 mg/m^2^) on day 1, or docetaxel (75 mg/m^2^) on day 1. No other antitumor treatments, such as surgery, targeted drugs, or radiotherapy were received before the chemotherapy.

Relevant clinical data were collected from patients’ medical records, including disease stage, gender, age, smoking history, tumor pathological examination results, treatment regimen, number of chemotherapy cycles, and dose. The efficacy of the chemotherapy was evaluated according to the response evaluation criteria in solid tumors (RECIST) version 1.1 [9], and patients were divided into two groups, with partial response (PR) classified as platinum–sensitive, and stable disease (SD) and progressive disease (PD) classified as platinum–resistant.

As shown in Figure 1A, this was a two cohort study. In the discovery cohort, 217 patients were enrolled; and excluded patients without an efficacy assessment (*n* = 15), failed DNA quality control (*n* = 9), and failed call rate < 90% (*n* = 1). In the validation cohort, 216 patients were enrolled; and excluded patients without an efficacy assessment (*n* = 14), failed DNA quality control (*n* = 7), and failed call rate < 90% (*n* = 3).

### 2.2. DNA Extraction, Genotyping, and Quality Control

Whole blood samples were collected from patients, and blood samples were preserved with EDTA disodium salt (EDTA-2NA). Genomic DNA was extracted from peripheral blood using the Wizard^®^ Genomic DNA Purification Kit (Promega, Madison, WI, USA) according to the instructions.

DNA samples (*n* = 227) from the discovery cohort were genotyped using Illumina BeadChip Array Global Screening Array-24+ V1.0 (Illumina Inc., San Diego, CA, USA). Quality control was performed on the original genotyping data by PLINK (v1.90b6.24) to filter unqualified samples and SNPs [10]. The data quality control was divided into SNP quality control and sample quality control. SNP quality control included (1) SNP call rate ≥ 90%; (2) the Hardy–Weinberg equilibrium test (HWE) ≥ 1 × 10^−5^; (3) minor allele frequency (MAF) ≥ 0.01; (4) mapped on autosomal chromosomes. Sample quality control included (1) sample call rates ≥ 90%; (2) heterozygosity check; (3) sex check; (4) sample relatedness check; (5) PCA analysis. Principal component analysis (PCA) was performed using the PLINK package. PLINK calculated the PC values from the SNP information of the samples, determined the location of the samples based on the PC1 and PC2 of each sample, and presented these in a scatter plot (Figure 1B). There are no stray samples in Figure 1B, so all 217 samples were included in the analysis. We used the impute2 software for imputation (INFO = 0.4). After imputation, another quality control was performed on the data before association analysis: (1) SNP call rate ≥ 90%; (2) MAF ≥ 0.01; (3) HWE ≥ 1 × 10^−5^; (4) sample call rate ≥ 90%; (5) sample relatedness check; (6) heterozygosity checks; (7) sex check. After quality control, there were 217 samples in our discovery cohort; a total of 104 in the sensitive group and 113 in the resistant group. The Hardy–Weinberg equilibrium test was failed by 2035 SNPs, and 151,215 SNPs were filtered out due to minor allele frequencies < 1%; a total of 629,4406 SNPs were included in the analysis.

In the validation cohort, SNP genotyping was performed using the Sequenom MassARRAY system (Sequenom, San Diego, CA, USA) according to the manufacturer’s instructions. The quality control criteria included (1) sample call rates ≥ 90%; (2) SNP call rate ≥ 90%; (3) MAF ≥ 0.01; (4) HWE ≥ 0.05; (5) sample relatedness check; (6) heterozygosity check; (7) sex check. The overall methodological flow is shown in Figure 2.

### 2.3. Genome–Wide Association Analysis and Linkage Disequilibrium (LD) Pruning

Multivariate logistic regression analysis was used, adjusting for pathology type, disease stage, age, smoking history, chemotherapy regimen, and sex as covariates, and performed genome–wide association analysis under an additive model. LD pruning was carried out for all SNPs, with a window size of 500 KB, a step size of 50ct, and the r2 threshold for determining association set to 0.2.

### 2.4. Model Construction

SNPs with specific *p*-value intercepts were selected after association analysis using multivariate logistic regression in PLINK. In choosing these SNPs, the PLINK package was used to calculate all the odds ratio (OR) values of the SNPs. Models were built using different algorithms: PRS, BN, NB, RF, LR, SVM, Bagging, KNN, and RT. The PRS score was calculated by the PRSice software (PRSice, V2.3.5) [11]. After obtaining the PRS score of each sample, logistic regression was used to build the model with the SPSS 25.0 software (SPSS Inc, Chicago, IL, USA). For other algorithms, SNPs were assigned to 0, 1, and 2 according to the genotyping results of each SNP in the patient. Using BayesNet, NaiveBayes, RF, Logistic, SVM, Bagging, KNN, and Random Tree (RT) to build the models was realized by the Knowledge Analysis (WEKA) software (V3.8.5) [12]. AU-ROC, specificity, sensitivity (recall rate), and accuracy were used to evaluate and compare the overall performance of the models.

Then, the SNPs selected from the discovery cohort were genotyped in the validation cohort to verify the model. Finally, clinical factors were integrated into the model to observe whether we can have a better prediction effect.

## 3. Results

### 3.1. Characteristics of the Study Population

The study was divided into two cohorts: the discovery cohort and the validation cohort. The discovery cohort was used to select SNPs to build the model, and the validation cohort was used to verify the contribution of the selected SNPs to the phenotype in another independent cohort. The sample numbers of the case group and the control group were roughly the same. There were 217 patients in the discovery cohort, 104 (47.93%) were evaluated as sensitive and 113 (52.07%) as resistant to platinum chemotherapy; and 216 patients in the validation cohort, 102 (47.22%) were evaluated as sensitive, and 114 (52.78%) as resistant. The pathology type, disease stage, age, smoking history, chemotherapy regimen, and gender of the patients were collected. Table 1 lists the clinical and pathological characteristics of the patients in both groups. It can be seen that clinical factors had no statistically significant influence on phenotypes (*p* ≥ 0.05). In addition, the proportion of each clinical phenotype subtype is consistent between the discovery and validation cohorts.

### 3.2. GWAS to Identify SNPs Associated with Platinum–Based Chemotherapy Response

To find SNPs that have an impact on the platinum chemotherapy effect, GWAS was adopted to select the SNPs in the discovery cohort. To correct the covariates, logistic regression was used for an association analysis. Through a series of SNP and sample quality controls, 217 individuals (113 cases and 104 controls) were included in the analysis, a total of 6,294,406 SNPs were included in the analysis, and 6,278,311 SNPs obtained the determined *p*-value after association analysis. The Manhattan plot (Figure 3) shows the results of the genome–wide association analysis.

The SNPs with significant phenotypic contributions were selected to establish the prediction models. At first, the SNPs with *p* values < 10^−3^ were analyzed. LD exists when alleles from two adjacent genetic variants co-occur in a non-random, linkage manner. To eliminate the false positives caused by the inclusion of SNPs with LD, we conducted linkage imbalance pruning for SNPs after a logistic multiple Correction Association analysis. Table 2 showed the P and OR values of all SNPs with *p* < 10^−3^ after the logistic multiple correction association analysis and LD pruning, as well as the genes and mutation types of SNPs. There were 57 SNPs with *p* < 10^−3^, of which 59.65% of the SNPs were located on genes and 40.35% on gene deserts, and most of the SNPs on genes were introns. Considering that there were still many SNPs, the SNPs with *p* < 10^−4^ were selected. There were four SNPs with *p* < 10^−4^: rs7463048 (*p* = 4.60 × 10^−5^, OR = 2.452), rs17176196 (*p* = 6.16 × 10^−5^, OR = 0.3409), rs527646 (*p* = 6.96 × 10^−5^, OR = 0.3438), and rs11134542 (*p* = 9.89 × 10^−5^, OR = 2.441); all four of these SNPs are located on genes. rs7463048 is an intron of *LOC105375676*, rs17176196 is an intron of the read–through gene ANKRD34C antisense RNA 1 (*ANKRD34C-AS1)*, rs527646 is an intron of opioid binding protein/cell adhesion molecule like (*OPCML)*, and rs11134542 is an intron of slit guidance ligand 3 (*SLIT3*) or a non-coding mutation of *LOC105377713*.

At the same time, to compare our new model with the previous model, 20 SNPs from the previous model were analyzed. Of these, two SNPs were not among the SNPs we analyzed, and five SNPs failed to pass the quality control. Thirteen SNPs remained after LD pruning.

In general, different numbers of SNPs were selected to build the models. The SNPs used for modeling were divided into three groups: (1) SNPs with *p* < 10^−3^; (2) SNPs with *p* < 10^−4^; (3) SNPs that were selected by the candidate gene method in the previous model.

### 3.3. Construction of the Genetic Prediction Models in the Discovery Cohort

After selecting these three groups of SNPs, the SNPs were used to build the prediction models. The models established by these SNPs were named as, (1) medium correlation model (SNPs with *p* < 10^−3^); (2) high correlation model (SNPs with *p* < 10^−4^); (3) the prior model (SNPs selected by the candidate gene method in the previous model). To obtain the best prediction model, multiple algorithms were used to build the models. Eleven algorithms including PRS, BayesNet, NaiveBayes, RF, LR, SVM, Bagging, KNN, RT, DT, and AdaBoost were used. The sensitivity, specificity, accuracy, and the AU-ROC of each model were analyzed, as shown in Figure 4A–C and Table 3. Because accuracy integrates both specificity and sensitivity information, therefore, with these three indicators we are mainly referring to accuracy.

Among the models built with the 11 algorithms, the models built using PRS, BayesNet, NaiveBayes, LR, RF, and SVM have higher values of each indicator, and the prediction model built using PRS has the highest accuracy. In the medium correlation models, the model built with PRS has the best prediction effect (sensitivity 0.973; specificity 0.971; accuracy 0.972). In the high correlation models, PRS is also the best method (sensitivity 0.726; specificity 0.731; accuracy 0.728). In the prior models, the effect of PRS is much better than other algorithms. Only the model built using PRS obtained an accuracy above 0.6 (sensitivity 0.743; specificity 0.538; accuracy 0.645).

The results show that the medium correlation models with the PRS algorithm have the best performance. Among all of the model indicators, AU-ROC is one of the most commonly used indicators in evaluating binary classifiers, and shows the true positive rate relative to the false positive rate. Therefore, finally, AU-ROC was used to evaluate the performances of the models. Figure 4D shows the ROC curves of the best prediction models for the three groups of SNPs in the discovery cohort. The medium correlation model has the highest AUC (0.995), followed by the high correlation model (0.795), and the prior model has the lowest AUC (0.593).

### 3.4. Validation of the Genetic Prediction Models

Because in the discovery cohort, the medium correlation models and the high correlation models have a higher AUC than the prior models, therefore, only the groups of SNPs selected using GWAS were verified in another independent cohort. Again, since the models built using PRS have the highest accuracy among all the models, only PRS was used for modeling in the validation cohort. After quality control, 216 samples were included in our validation cohort. Figure 5A shows the ROC curves of the models. The results show that the high correlation model has the highest AUC in the validation cohort (AUC = 0.642), followed by the medium correlation model (AUC = 0.582).

Therefore, our model considered including four SNPs with *p* < 10^−4^ and adopted the PRS algorithm. By calculating the ORs and genotypes of the SNPs included in the analysis, PRS gives each sample a score; based on the scores, the samples were divided into three groups: low, medium, and high. Figure 5B shows the efficacy of chemotherapy in these three groups. In the low risk group, there were more platinum chemotherapy–sensitive patients; in the high risk group, there were more resistant patients.

### 3.5. Integrating Genetic and Clinical Factors to Further Improve the Performance of the Model

To further improve the model performance, clinical factors were included in the model [13,14,15]. The disease stage, pathology, age, gender, smoking history, and adjuvant chemotherapy regimens were included in the modeling, and the results are shown in Table 4 and Figure 6A. Among them, the AUC of the model can reach more than 0.67 by considering the genetic factor plus pathology (AUC = 0.67), or the genetic factor plus chemotherapy regimens (AUC = 0.673). Except for the pathology and chemotherapy regimens, inclusion of the other clinical factors did not significantly enhance model performance. Since the addition of pathology or chemotherapy regimens separately can improve the AUC of the model, both pathology and chemotherapy regimens were included in the model and the model’s performance was compared with the model that included all six clinical factors. The model performance was not significantly different for these two models (0.675 vs. 0.679). Therefore, the pathology and chemotherapy regimens were added to the model. In addition, we performed subgroup analyses to analyze the performance of genetic and clinical factors in the adenocarcinoma samples (110 samples) and the squamous carcinoma samples (101 samples) (Table 4). As Figure 6B,C shows, the performance of the model was slightly higher in adenocarcinoma than in squamous carcinoma, without significant differences.

After selecting SNPs to build the model and verify it, our model finally adopted PRS and included the four SNPs with *p* < 10^−4^ as well as the two clinical factors, pathology and chemotherapy regimen. By combining the discovery and validation cohorts to build this prediction model, the model had a sensitivity of 0.705, a specificity of 0.670, and an accuracy of 0.689, with an AUC of 0.726. Figure 7 shows the ROC of the model.

## 4. Discussion

In our study, four SNPs were identified that influence the response to platinum–based chemotherapy in NSCLC: rs7463048, rs17176196, rs527646, and rs11134542. rs527646 is an intron mutation of Opioid-binding Protein/Cell-adhesion Molecule-Like (*OPCML*), and rs11134542 is an intron mutation of *SLIT3*. The four SNPs were included in the model along with two clinical factors, pathology and chemotherapy regimen, modeled with PRS, and the AUC of the NSCLC platinum drugs response prediction model was 0.726.

*OPCML* is a potent tumor suppressor exposed on the cell surface, to which it is anchored by a glycosylphosphatidylinositol (GPI) group. *OPCML* functions as a tumor suppressor and is silenced in over 80% of ovarian cancers by loss of heterozygosity and by epigenetic mechanisms. In addition to ovarian cancer, *OPCML* is also hypermethylated in other solid tumors such as cervical cancer, lung, brain, live, bladder, prostate cancer, colorectal and gastric cancer, and lymphoma. *OPCML* exerts its tumor suppressor effect by inhibiting several cancer hallmark phenotypes in vitro, and abrogating tumorigenesis in vivo, by downregulating/inactivating a specific spectrum of Receptor Tyrosine Kinases (*RTKs*), including *EphA2*, *FGFR1*, *FGFR3*, *HER2*, *HER4*, and *AXL* [16,17,18,19,20,21,22,23].

*SLIT3* is also associated with tumor proliferation, migration, and invasion. The Slit/Robo signaling pathway is reportedly involved in breast cancer development and metastasis. Overexpression of Slit/Robo induces its tumor suppressive effects possibly by altering β-catenin/E-cadherin-mediated cell-cell adhesion in breast cancer cells [24,25].

Our study suggests that, when discussing complex phenotypes with many influencing factors, the contribution of a single SNP is limited, and the predictive effect could be better if multiple SNPs with higher contributions were included in the model [26]. Through high-throughput screening, more than one SNP was associated with the efficacy of NSCLC platinum chemotherapy. In this study, the most significant SNP has a small impact on the efficacy of NSCLC platinum chemotherapy, with an OR value of less than 2.5. However, a better predictive effect can be achieved by including multiple relatively significant SNPs into the model at the same time.

We also suggest that PRS is better than other algorithms at building predictive models. PRS is the mainstream method to discuss the influence of genetic factors on phenotypes in recent years, and has become the standard to quantify the genetic role in disease risk prediction, such as the prediction of the risk of inflammatory bowel disease (IBD), thyroid cancer, lung cancer, fatal prostate cancer, colorectal cancer, and type 2 diabetes [27,28,29,30,31,32]. PRS can be used to predict a variety of cancer susceptibilities, but its role in the cancer field may not be limited to cancer risk. Our results show that PRS can be used not only to quantify the prediction of disease risk but also to predict drug efficacy, with superior results to traditional data mining methods. This provides a new idea for the selection of modeling methods for evaluating drug efficacy.

Our team has been focused on the pharmacogenomics of platinum responses for over a decade. A number of genes and mutations were observed that affect the efficacy of platinums. For example, our study found that rs2280496 and rs189178649 in the ADCY1 gene were associated with the sensitivity of platinum–based chemotherapy in NSCLC patients [33]. It was also found that the eIF3A R803K somatic mutation has the potential to predict chemotherapy resistance in SCLC [34]. These studies looked at the effect of mutations in a single gene on the response to platinum. As mentioned above, in order to further understand the influence of genetic factors on the response to platinum–based chemotherapy, we established a prediction model for NSCLC platinum–based chemotherapy efficacy based on the candidate gene method [8]. In this study, a large part of the content is to compare the new model with the prior model. The specificity of the model has been greatly improved. In addition, there are thirteen SNPs selected for modeling by the candidate gene method, while only four SNPs are included in the new model, indicating that the SNPs included in the new model are more representative. The genotypes of rs7463048, rs17176196, rs527646, and rs11134542 in patients with NSCLC can predict the efficacy of platinum–based drugs and thus guide clinical use.

When discussing clinical factors, we suggest that most of the clinical factors had no significant effect on the model’s performance (disease stage, age, gender, smoking history), and the improvement of the model’s performance when only considering pathology and chemotherapy regimen was similar to that when considering all clinical factors.

Therefore, SNPs were selected for the first time by the GWAS method to build a predictive model for platinum drug efficacy in NSCLC, modelled by the PRS method, and two clinical factors with significant effects on phenotype were incorporated into the model, ultimately perfecting the previous predictive model for platinum drug efficacy in NSCLC.

Many studies have attempted to integrate genetic factors with clinical characteristics to develop models for predicting responses to drug therapies. Most of these models have an AUC above 0.7. Meenal Gupta et al. identified a predictive model for response to atypical antipsychotic monotherapy treatment in south Indian schizophrenia patients, their model had an AUC of 0.733 [35]. Kimi Drobin et al. found circulating proteins and a SNP variant of VEGFA predict radiosensitivity in breast cancer (AUC = 0.76) and head-and-neck cancer (AUC = 0.89) [36]. In contrast to other studies, the main shortcoming of our model is that the AUC should be further improved. Subsequently, the sample size needs to be expanded to find SNPs with greater contributions to the phenotype for modeling.

In conclusion, by GWAS, we identified four genetic susceptibility SNPs and two clinical factors affecting platinum–based chemosensitivity in NSCLC, and constructed a new predictive model.

## Figures and Tables

**Figure 1 jcm-12-01318-f001:**
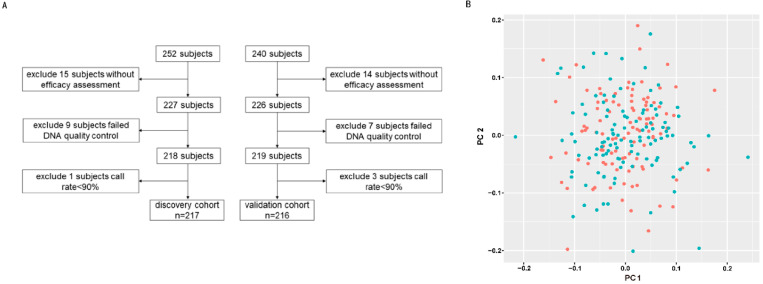
Sample screening and quality control. (**A**) Sample filtering for discovery and validation cohorts. (**B**) The PCA of the discovery cohort. The red spots represent sensitive patients and the green spots represent resistant patients. The horizontal and vertical coordinates are the PC1 and PC2 values for each sample.

**Figure 2 jcm-12-01318-f002:**
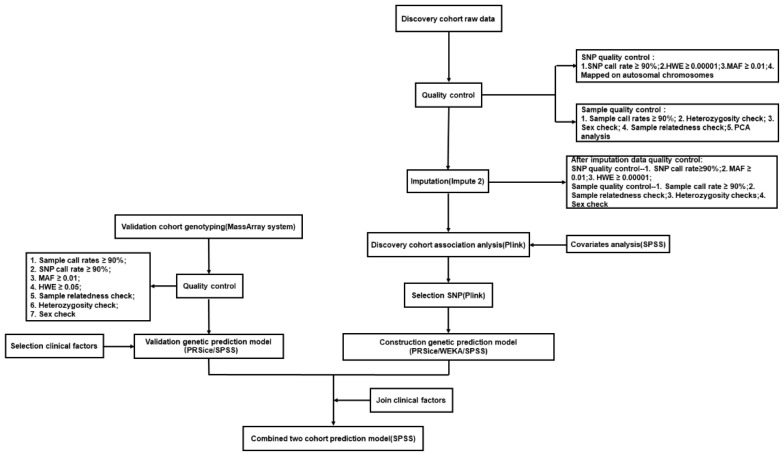
Methodological flow chart of the whole study.

**Figure 3 jcm-12-01318-f003:**
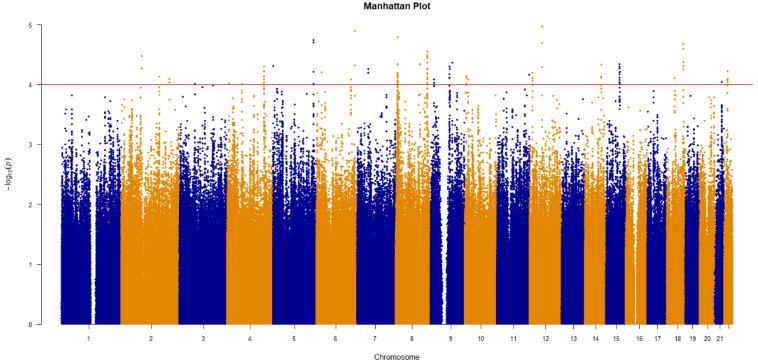
The result of genome–wide association analysis for the discovery cohort. The *p*-value for correlation is expressed as −log10(*p*). *p*-values were calculated from multivariate logistic regression analysis.

**Figure 4 jcm-12-01318-f004:**
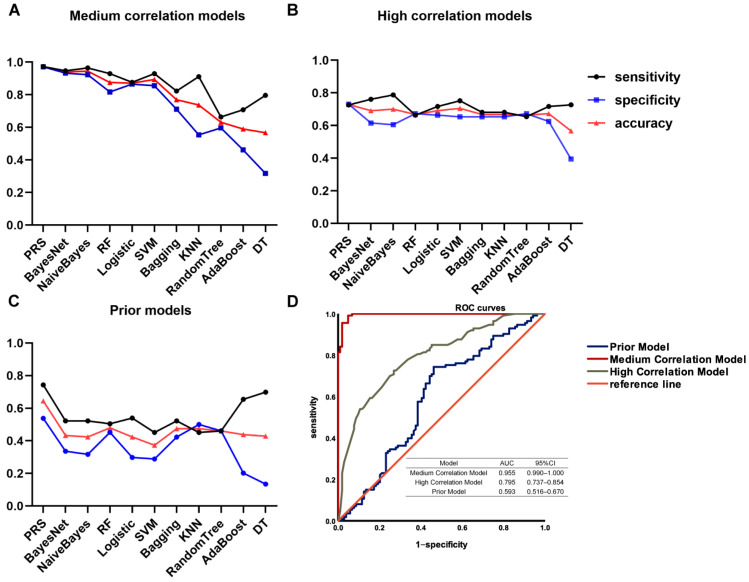
The performance of the platinum–based chemotherapy response genetic model in the discovery cohort. The sensitivity (black curve), specificity (blue curve), accuracy (red curve), and AUCs of ROC curves of the models established using different algorithms were indicated. (**A**) The medium correlation models; (**B**) the high correlation models; (**C**) the prior models; (**D**) AUCs of ROC curves of the models established by PRS.

**Figure 5 jcm-12-01318-f005:**
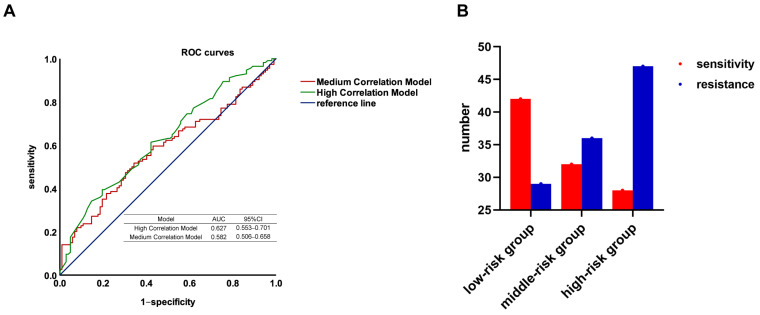
The performance of the platinum–based chemotherapy response genetic model in the validation cohort. (**A**) AUCs of ROC curve of models established by PRS; (**B**) Using PRS to group the validation cohort. Based on the scores, the samples are divided into three groups: low, medium, and high. The number of sensitive (red) and resistant (blue) patients in each group is shown.

**Figure 6 jcm-12-01318-f006:**
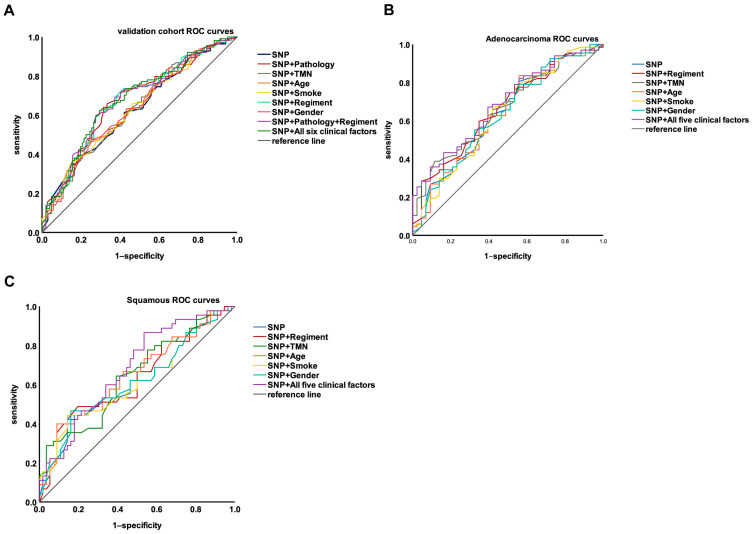
(**A**) The performance of the platinum–based chemotherapy response models established by SNPs combined with different clinical factors in the validation cohort. (**B**) The performance of the platinum–based chemotherapy response models established by SNPs combined with different clinical factors in the validation cohort of adenocarcinoma samples. (**C**) The performance of the platinum–based chemotherapy response models established by SNPs combined with different clinical factors in the validation cohort of squamous carcinoma samples.

**Figure 7 jcm-12-01318-f007:**
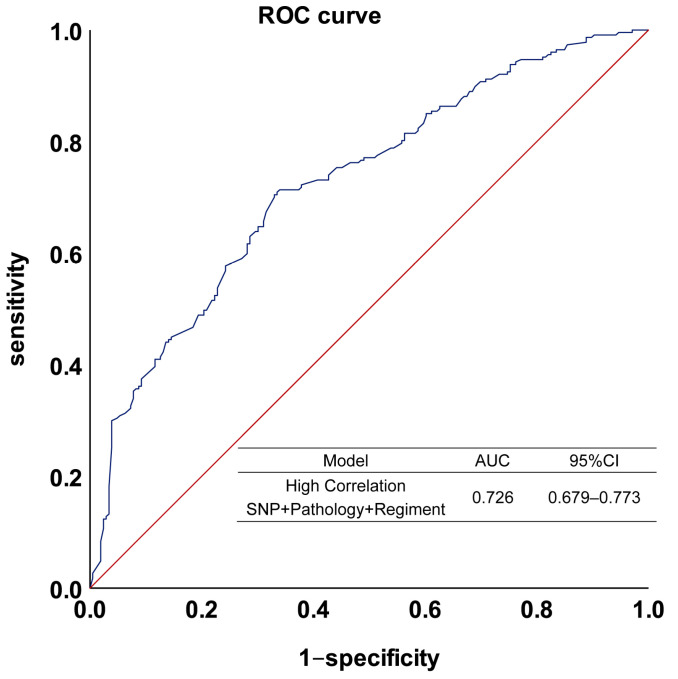
The performance of the platinum–based chemotherapy response model in the combined cohort. The model was established by SNPs combined with pathology and regimen. The blue line is the model fit curve and the red line is the reference line.

**Table 1 jcm-12-01318-t001:** Characteristics of NSCLC patients for the two cohorts.

Characteristics	Discovery Stage			Validation Stage		
	Responders (%) (*n* = 104)	Non-Responders (%) (*n* = 113)	*p* Value	Responders (%) (*n* = 102)	Non-Responders (%) (*n* = 114)	*p* Value
Histology						
Adenocarcinoma	51 (49.04)	56 (49.56)	0.74	43 (42.16)	67 (58.77)	0.05
Squamous	51 (49.04)	54 (47.79)		56 (54.90)	45 (39.47)	
Other	2 (1.92)	3 (2.65)		3 (2.94)	2 (1.76)	
Stage						
I	NA	2 (1.77)	0.72	1 (0.98)	NA	0.40
II	1 (0.96)	1 (0.88)		3 (2.94)	1 (0.88)	
III	32 (30.77)	33 (29.20)		36 (35.29)	36 (31.58)	
IV	71 (68.27)	77 (68.15)		62 (60.79)	77 (67.54)	
age						
≤55	44 (42.31)	50 (44.25)	0.22	43 (42.16)	52 (45.61)	0.16
>55	60 (57.69)	63 (55.75)		59 (57.84)	62 (54.39)	
Smoking status						
Nonsmoker	38 (36.54)	36 (31.86)	0.66	27 (26.47)	40 (35.09)	0.17
Smoker	66 (63.46)	77 (68.14)		75 (73.53)	74 (64.91)	
Sex						
Male	76 (73.08)	94 (83.19)	0.41	84 (82.35)	89 (78.07)	0.43
Female	28 (26.92)	19 (16.81)		18 (17.65)	25 (21.93)	
Chemotherapeutic regimens						
platinum/pemetrexed	42 (40.39)	50 (44.25)	0.84	38 (37.25)	61 (53.51)	0.11
platinum/gemcitabine	40 (38.46)	44 (38.94)		38 (37.25)	34 (29.82)	
platinum/paclitaxel	15 (14.42)	12 (10.62)		18 (17.65)	14 (12.28)	
platinum/docetaxel	7 (6.73)	7 (6.19)		8 (7.85)	5 (4.39)	

**Table 2 jcm-12-01318-t002:** The SNPs with *p* < 10^−3^ associated with platinum–based chemotherapy response in the discovery cohort.

SNP	CHR	OR	*p*	Gene	Mutation Type
rs7463048	8	2.45	4.60 × 10^−5^	*LOC105375676*	Intron
rs17176196	15	0.34	6.16 × 10^−5^	*ANKRD34C-AS1*	Intron
rs527646	11	0.34	6.96 × 10^−5^	*OPCML*	Intron
rs11134542	5	2.44	9.89 × 10^−5^	*SLIT3, LOC105377713*	Intron, Non-Coding Transcription
rs10078147	5	2.15	1.32 × 10^−4^	*SLC1A3*	Intron
rs9378820	6	2.32	1.39 × 10^−4^	NA	NA
rs7113019	11	2.40	1.52 × 10^−4^	*JHY*	Intron
rs7005628	8	0.31	1.76 × 10^−4^	NA	NA
rs11617804	13	3.68	1.77 × 10^−4^	*FAM155A*	Intron
rs7005216	8	0.32	1.79 × 10^−4^	NA	NA
rs944927	20	0.42	1.83 × 10^−4^	*LOC105372656*	Intron
rs11808688	1	0.44	1.91 × 10^−4^	*ATP2B4*	Intron
rs17121520	14	0.12	2.02 × 10^−4^	NA	NA
rs1620779	11	2.23	2.33 × 10^−4^	*LINC02698*	Intron
rs60976228	6	0.24	2.37 × 10^−4^	*FANCE*	Intron
rs9496862	6	0.32	2.67 × 10^−4^	*LOC105378036*	2 KB Upstream
rs7225086	17	2.21	2.89 × 10^−4^	*RTN4RL1*	Intron
rs10859720	12	2.63	3.97 × 10^−4^	*LOC102724960*	Intron
rs949561	3	2.37	3.99 × 10^−4^	*LOC107986169*	Intron
rs12545542	8	0.47	4.78 × 10^−4^	*LOC100128993*	Intron
rs150131032	1	0.45	4.88 × 10^−4^	*LINC01344*	Intron
rs4146476	4	0.49	5.31 × 10^−4^	NA	NA
rs76581411	14	0.22	5.69 × 10^−4^	NA	NA
rs7313678	12	2.12	5.79 × 10^−4^	NA	NA
rs2060515	4	2.26	6.31 × 10^−4^	NA	NA
rs7134969	12	2.16	6.31 × 10^−4^	*LOC105370003*	Intron
rs1601345	14	0.48	6.54 × 10^−4^	NA	NA
rs3762678	3	0.22	6.56 × 10^−4^	*ACAD11, NPHP3-ACAD11*	Intron
rs6873965	5	2.15	6.76 × 10^−4^	NA	NA
rs9832471	3	0.27	6.78 × 10^−4^	NA	NA
rs1080178	2	0.47	6.94 × 10^−4^	*UPP2*	Intron
rs4244459	8	1.92	7.06 × 10^−4^	*LOC105379311*	Intron
rs77176301	3	0.48	7.07 × 10^−4^	NA	NA
rs12962513	18	2.35	7.18 × 10^−4^	NA	NA
exm2267842	15	0.46	7.46 × 10^−4^	*ANP32A, ANP32A-IT1*	Intron, 2KB Upstream
rs4520608	11	2.14	7.51 × 10^−4^	*OPCML*	Intron
rs77859697	12	0.30	8.01 × 10^−4^	*LINC02388, LOC100506869*	Intron
rs1523483	3	0.49	8.02 × 10^−4^	NA	NA
rs145303018	9	0.35	8.10 × 10^−4^	NA	NA
rs149864625	10	0.21	8.12 × 10^−4^	*LRMDA*	Intron
rs1425351	4	2.91	8.21 × 10^−4^	*COL25A1*	Intron
rs74531987	22	2.51	8.26 × 10^−4^	NA	NA
rs201395	12	0.47	8.45 × 10^−4^	NA	NA
rs4518067	3	0.46	8.47 × 10^−4^	NA	NA
rs1472094	4	0.18	8.56 × 10^−4^	*LOC105374528*	Intron
rs56845228	7	0.27	8.70 × 10^−4^	*LOC105375523*	Intron
rs11957972	5	0.15	8.99 × 10^−4^	NA	NA
rs7019568	9	0.31	9.07 × 10^−4^	NA	NA
rs7864626	9	0.48	9.08 × 10^−4^	*GNA14, GNA14-AS1*	Intron
rs244046	4	2.15	9.15 × 10^−4^	NA	NA
rs139740488	5	2.08	9.18 × 10^−4^	NA	NA
rs76443044	20	0.22	9.37 × 10^−4^	*CDH4*	Intron
rs2050346	10	0.35	9.48 × 10^−4^	*LOC105376391*	Intron
rs1860139	14	0.40	9.48 × 10^−4^	*LINC02274*	Non-Coding Transcription
rs2324596	13	0.49	9.71 × 10^−4^	NA	NA
rs9285510	6	3.03	9.93 × 10^−4^	*STXBP5-AS1*	Intron Variant
rs28517685	19	3.01	9.97 × 10^−4^	*LOC105372349, LOC100420587*	Non-Coding Transcription, Intron

**Table 3 jcm-12-01318-t003:** (A) The sensitivity, specificity, accuracy, and the AU-ROC of each model established by the SNPs with *p* < 10^−3^ in the discovery cohort. (B) The sensitivity, specificity, accuracy, and the AU-ROC of each model established by the SNPs with *p* < 10^−4^ in the discovery cohort. (C) The sensitivity, specificity, accuracy, and the AU-ROC of each model established by the SNPs selected using the candidate gene method in the discovery cohort.

(A)				
**Methods**	**Sensitivity**	**Specificity**	**Accuracy**	**AUC**
PRS	0.97	0.97	0.97	0.995
BayesNet	0.95	0.93	0.94	0.988
NaiveBayes	0.96	0.92	0.94	0.988
RF	0.93	0.82	0.88	0.956
Logistic	0.88	0.87	0.87	0.946
SVM	0.93	0.86	0.89	0.893
KNN (IBK)	0.91	0.55	0.74	0.750
Bagging	0.82	0.71	0.77	0.859
Random Tree	0.66	0.60	0.63	0.612
DT	0.80	0.32	0.57	0.551
AdaBoost	0.71	0.46	0.59	0.673
(B)				
**Methods**	**Sensitivity**	**Specificity**	**Accuracy**	**AUC**
PRS	0.73	0.73	0.73	0.795
BayesNet	0.76	0.62	0.69	0.773
NaiveBayes	0.79	0.61	0.70	0.776
RF	0.66	0.67	0.69	0.712
Logistic	0.72	0.66	0.69	0.761
SVM	0.75	0.65	0.71	0.702
KNN (IBK)	0.68	0.65	0.67	0.715
Bagging	0.68	0.65	0.67	0.724
Random Tree	0.65	0.67	0.66	0.682
DT	0.73	0.39	0.57	0.525
AB	0.72	0.63	0.67	0.756
(C)				
**Methods**	**Sensitivity**	**Specificity**	**Accuracy**	**AUC**
PRS	0.74	0.54	0.64	0.593
BayesNet	0.52	0.34	0.43	0.406
NaiveBayes	0.52	0.32	0.42	0.402
RF	0.50	0.45	0.48	0.490
Logistic	0.54	0.30	0.42	0.397
SVM	0.45	0.29	0.37	0.370
KNN (IBK)	0.45	0.50	0.47	0.485
Bagging	0.52	0.42	0.47	0.472
Random Tree	0.46	0.46	0.46	0.492
DT	0.70	0.13	0.43	0.380
AB	0.65	0.20	0.44	0.419

**Table 4 jcm-12-01318-t004:** (A) The AUCs of models established by SNPs combined with different clinical factors in the discovery cohort. (B) The AUCs of models established by SNPs combined with different clinical factors in the discovery cohort of adenocarcinoma samples. (C) The AUCs of models established by SNPs combined with different clinical factors in the discovery cohort of squamous carcinoma samples.

(A)		
**Model**	**AUC**	**95% CI**
SNP	0.627	0.553–0.701
SNP + Regimen	0.673	0.602–0.745
SNP + Pathology	0.670	0.597–0.742
SNP + TMN	0.635	0.562–0.709
SNP + Age	0.637	0.563–0.710
SNP + Smoke	0.633	0.559–0.706
SNP + Gender	0.633	0.559–0.707
SNP + Pathology + Regimen	0.675	0.603–0.746
SNP + Regimen + Pathology + TMN + Age + Smoke + Gender	0.679	0.608–0.750
(B)		
**Model**	**AUC**	**95% CI**
SNP	0.638	0.531–0.746
SNP + Regimen	0.661	0.558–0.764
SNP + TMN	0.659	0.557–0.761
SNP + Age	0.634	0.526–0.741
SNP + Smoke	0.64	0.532–0.747
SNP + Gender	0.635	0.527–0.742
SNP + Regimen + TMN + Age + Smoke + Gender	0.685	0.585–0.784
(C)		
**Model**	**AUC**	**95% CI**
SNP	0.612	0.499–0.724
SNP + Regimen	0.632	0.521–0.743
SNP + TMN	0.646	0.537–0.754
SNP + Age	0.648	0.538–0.757
SNP + Smoke	0.614	0.501–0.727
SNP + Gender	0.617	0.505–0.729
SNP + Regimen + TMN + Age + Smoke + Gender	0.681	0.577–0.785

## Data Availability

Datasets of the current study are not publicly available but are available from the corresponding author on reasonable request.

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
