# Peer review of "Establishing a Prediction Model for the Efficacy of Platinum—Based Chemotherapy in NSCLC Based on a Two Cohorts GWAS Study"

_jcm, 2023, doi:10.3390/jcm12041318_

Round 1
Reviewer 1 Report
1. Though it is a very interesting topic with potential implication of clinical factors, the results should be presented by histology type (at least squamus - non squamus) and disease stage, as the expected response to therapy differs significantly.
2. How were stages I-III included without any surgery or radiation therapy? This undertreatment as the patients did not receive the standard of care.
Reviewer 2 Report
The work by Xiao et al entitled "Establishing a prediction model for the efficacy of platinum-based chemotherapy in NSCLC based on a two cohorts GWAS study" developed a predictive model for efficacy of platinum-based chemotherapy in NSCLC. The study identified four SNPs and two clinical factors as important contributors and the implemented model achieved an AUC of 0.726.
1) Need to clarify what features were used to calculate the PCA in figure 1B.
2) Table 3, showed the performance of model based on PRS is highest, then why authors selected and report final model with low accuracy.
3) The author must provide the codes and model on GitHub, so that scientific community get benefitted with it.
Round 2
Reviewer 1 Report
Thank you for considering the suggestions of improvment. Nice work.
Author Response
Thanks very much for your suggestions on our manuscript. They are all valuable and very helpful for revising and improving our paper, as well as the important guiding significance to our research. We have studied comments carefully and have made correction. We hope that our revisions will meet with your approval.